# Renoprotective Effect of *Pediococcus acidilactici* GKA4 on Cisplatin-Induced Acute Kidney Injury by Mitigating Inflammation and Oxidative Stress and Regulating the MAPK, AMPK/SIRT1/NF-κB, and PI3K/AKT Pathways

**DOI:** 10.3390/nu14142877

**Published:** 2022-07-13

**Authors:** Wen-Hsin Lin, Wen-Ping Jiang, Chin-Chu Chen, Li-Ya Lee, You-Shan Tsai, Liang-Hsuan Chien, Ya-Ni Chou, Jeng-Shyan Deng, Guan-Jhong Huang

**Affiliations:** 1College of Pharmacy, China Medical University, Taichung 404, Taiwan; wslin@mail.cmu.edu.tw; 2Department of Pharmacy, Chia Nan University of Pharmacy and Science, Tainan 717, Taiwan; wpjiang@gm.cnu.edu.tw; 3Biotech Research Institute, Grape King Bio Ltd., Taoyuan 330, Taiwan; gkbioeng@grapeking.com.tw (C.-C.C.); ly.lee@grapeking.com.tw (L.-Y.L.); you.shan.tsai@gmail.com (Y.-S.T.); 4Department of Chinese Pharmaceutical Sciences and Chinese Medicine Resources, College of Chinese Medicine, China Medical University, Taichung 404, Taiwan; u107047001@cmu.edu.tw (L.-H.C.); yanichoucmu@gmail.com (Y.-N.C.); 5Department of Food Nutrition and Healthy Biotechnology, Asia University, Taichung 413, Taiwan

**Keywords:** *Pediococcus acidilactici* GKA4, cisplatin, acute kidney injury, anti-inflammation, oxidative stress

## Abstract

Acute kidney injury (AKI) describes a sudden loss of kidney function and is associated with a high mortality. *Pediococcus acidilactici* is a potent producer of bacteriocin and inhibits the growth of pathogens during fermentation and food storage; it has been used in the food industry for many years. In this study, the potential of *P. acidilactici* GKA4 (GKA4) to ameliorate AKI was investigated using a cisplatin-induced animal model. First, mice were given oral GKA4 for ten days and intraperitoneally injected with cisplatin on the seventh day to create an AKI mode. GKA4 attenuated renal histopathological alterations, serum biomarkers, the levels of inflammatory mediators, and lipid oxidation in cisplatin-induced nephrotoxicity. Moreover, GKA4 significantly decreased the expression of inflammation-related proteins and mitogen-activated protein kinase (MAPK) in kidney tissues. Eventually, GKA4 also increased the levels of related antioxidant enzymes and pathways. Consistently, sirtuin 1 (SIRT1) upregulated the level of autophagy-related proteins (LC3B, p62, and Beclin1). Further studies are needed to check our results and advance our knowledge of the mechanism whereby PI3K inhibition (wortmannin) reverses the effect of GKA4 on cisplatin-treated AKI. Taken together, GKA4 provides a therapeutic target with promising clinical potential after cisplatin treatment by reducing oxidative stress and inflammation via the MAPK, AMP-activated protein kinase (AMPK)/SIRT1/nuclear factor kappa B (NF-κB), and phosphatidylinositol 3-kinase (PI3K)/protein kinase B (AKT) axes.

## 1. Introduction

Cisplatin (cis-diamminedichloroplatinum II), an inorganic platinum derivative, is used to treat a variety of tumors in the bladder, testis, and ovary, as well as non-small-cell lung cancer [1]. Nevertheless, cisplatin or its metabolites can be reabsorbed in the renal tubules, which not only kills malignant tumors but also causes severe damage to human normal tissue cells, resulting in peripheral neuropathy, allergy, and nephrotoxicity, followed by tubular cell death and acute kidney injury (AKI), with long-term side effects and risk of death [2]. Nephrotoxicity has been reported in approximately 25% of patients receiving cisplatin. Apart from supportive methods such as dialysis and renal replacement therapy, there are no specific treatment strategies for AKI in patients. AKI is characterized by the abrupt loss of renal function within hours and causative factors consist mainly of ischemia, nephrotoxic agents, and sepsis. The occurrence of AKI can lead to acute renal failure and cell death, peritubular endothelial dysfunction, and inflammatory cell infiltration. The regulation of cisplatin-treated AKI is unclear, and many studies have shown that renal reactive oxygen species (ROS) and inflammatory response contribute to AKI. Cisplatin accumulates in the kidney and induces cells to generate free radicals, thereby damaging renal tubular cells. Therefore, antioxidant defense and inflammation are the key markers of cisplatin-induced nephrotoxicity [3]. 

Oxidative stress can damage cellular structures that are involved in the progression of cisplatin-challenged kidney injury. ROS are generated under cisplatin treatment, which is relative to the expression of endogenous antioxidant enzymes [4,5]. Oxidative stress leads to renal tubular epithelial cell damage and the loss of renal function [6]. In addition, it leads to excessive cytokine production and secretion from infiltrating pro-inflammatory cells and renal tubular epithelial cells during cisplatin-induced AKI [6]. Therefore, the inhibition of tumor necrosis factor-α (TNF-α) can effectively protect mice after cisplatin-treated AKI [7]. In addition, renal tubular epithelial cells secrete various chemokines during cisplatin-induced AKI, so pro-inflammatory cells including neutrophils and macrophages infiltrate the damaged kidney [8]. The excessive accumulation of pro-inflammatory cells can cause tissue damage through cytokine and ROS production. In addition, oxidative stress activates the inflammatory axis to increase the level of nuclear factor (NF)-κB, the nuclear-factor-erythroid-2-related factor 2 (Nrf2)/heme oxygenase-1 (HO-1) axis, MAPK kinases, and other redox-sensitive signals [9].

Phosphoinositide-3-kinases (PI3K) are major regulators of cell growth and metabolism. Protein kinase B (AKT), a phosphorylation-activated kinase downstream of PI3K, is also important for targeting and controlling biosynthetic metabolic pathways including cell survival and proliferation, autophagy, and apoptosis [10]. All sirtuin families (SIRT1–7) share a common catalytic domain that binds NAD^+^. SIRT1 (sirtuin 1) is an NAD^+^-dependent deacetylase that plays key roles in biological responses, including aging, cancer, glucose metabolism, and energy homeostasis [11]. SIRT1 has been shown to reduce oxidative stress, inflammation, cellular senescence, and apoptosis [11]. However, it is currently unclear if the PI3K/AKT signaling pathway is involved in the development of AKI.

*Pediococcus acidilactici* is a Gram-positive coccus, facultative anaerobic fermentative lactic acid bacterium, that is usually found in pairs or tetrads. It is suitable for a wide range of growth conditions and, thus, readily colonizes the digestive tract [12]. *P. acidilactici* has been shown to produce bacteriocins that can colonize the human gut and, at the same time, have beneficial effects including adjusting the composition of gut bacteria, rebuilding the barrier function of the intestinal mucosa, and enhancing the defense capacity of the digestive tract [13,14,15]. However, there are currently few reports suggesting that probiotic products may have renoprotective effects. Thus, we illustrated the oral administration of the *P. acidilactici* GKA4 and evaluated its possible therapeutic applications in vivo. These data suggest that GKA4 can act as an acceptable probiotic target in cisplatin-challenged AKI. GKA4 may be a good complementary food for the prevention of AKI.

## 2. Materials and Methods

### 2.1. Preparation of Samples

*P. acidilactici* GKA4 isolated from fresh vegetables was isolated by Grape King Bio Ltd., Taoyuan, Taiwan [16]. The probiotic GKA4 strain was grown using MRS broth (BD Difco, Sparks, MD, USA) and then subcultured into 1.2 L MRS broth in 2 L flasks at 37 °C. The culture was then expanded into a 5-ton fermentor at 37 °C with a pH of 6.0 for 1 day using a medium consisting of glucose (5%), yeast extract (2.0%), MgSO_4_ (0.05%), and K_2_HPO_4_ (0.1%). For this study, liquid cultures were lyophilized to powder and then dissolved in 0.5% carboxymethyl cellulose (CMC) for administration. Oral probiotic powder had a cell count greater than or equal to 2 × 10^11^ colony forming units (CFU)/g. Amifostine (AMF) was also dissolved in 0.5% CMC.

### 2.2. Reagents

Cisplatin, AMF, wortmannin, and other reagents were acquired from Sigma-Aldrich (St. Louis, MO, USA). Creatinine (CRE) and blood urea nitrogen (BUN) detection kits were available from HUMAN Diagnostics Worldwide (Wiesbaden, Germany). The serum levels of cytokines IL-1β, IL-6, and TNF-α were detected with the ELISA MAX Deluxe Kit (BioLegend, San Diego, CA, USA). Antibodies against cyclooxygenase-2 (COX-2) (1:1500), p-JNK (1:1000), catalase (1:2000), superoxidase dismutase 1 (SOD1) (1:1000), SIRT1 (1:1000), AMPK (1:1000), glutathione peroxidase 3 (GPx3) (1:1000), and TLR-4 (1:1000) were provided from GeneTex (San Antonio, TX, USA). Antibodies against JNK (1:2000), p-ERK (1:500), ERK (1:1000), p-p38 (1:1000), p-CaMKK (1:1000), p-AMPK (1:1000), p-PI3K (1:1000), PI3K (1:2000), AKT (1:1500), p-AKT (1:1500), Beclin 1 (1:1000), LC3B (1:1000), P62 (1:1000), and p-IκB-α (1:1000) were bought from Cell Signaling Technology (Beverly, MA, USA). Antibodies that contrapose inducible NO synthase (iNOS) (1:1500), NF-κB (1:1000), IκBα (1:1500), HO-1 (1:2000), Nrf2 (1:1500), p38 (1:1000), and β-actin (1:10,000 dilution) were bought from Abcam (Cambridge, UK). β-actin was used as the loading control.

### 2.3. Animals

Eight-week-old ICR male mice (32 ± 3 g) were obtained from BioLASCO Taiwan Co., Ltd. Mice were placed on a 12 h light/dark cycle and a maintained temperature (23 °C) at 50% relative humidity for 3–5 days before the experiment. The entire experimental protocol was approved by the Animal Protection Committee of China Medical University (CMUIACUC-2020-327).

### 2.4. Research Design

Thirty male ICR mice were randomly divided into six groups (*n* = 5) including (1) the control group, which received an injection of saline (intraperitoneally: i.p.); (2) the cisplatin group, which received an injection of cisplatin (20 mg/kg, i.p.); (3) the amifostine (AMF; 200 mg/kg; i.p.) + cisplatin (20 mg/kg, i.p.) group; (4) the GKA4 (62.5 mg/kg; oral suspension) + cisplatin (20 mg/kg, i.p.) group; (5) the GKA4 (125 mg/kg; oral suspension) + cisplatin (20 mg/kg; i.p.) group; and (6) the GKA4 (250 mg/kg; oral suspension) + cisplatin (20 mg/kg; i.p.) group. The mice were administered GKA4 by oral gavage at doses of 62.5, 125, and 250 mg/kg once daily for 10 days (7 days before and 3 days after cisplatin injection). The control mice were orally administered saline daily. On day 7, AKI was induced in mice by an intraperitoneal injection of cisplatin (20 mg/kg) in the cisplatin and GKA4 groups. Mice were anaesthetized at 72 h after cisplatin injection to collect blood samples for measuring the serum biomarker, which were stored at −20 °C. Kidneys were collected at sacrifice for subsequent analysis. During this period, the body weights were measured weekly and averaged.

To assess the wortmannin in controlling cisplatin-treated AKI, twenty-five mice were randomly divided into five groups (*n* = 5) including (1) the control group (saline; i.p.), (2) the cisplatin group (20 mg/kg; i.p.), (3) the cisplatin (20 mg/kg; i.p.) + wortmannin (1.4 mg/kg; i.p.) group; (4) the cisplatin (20 mg/kg; i.p.) + GKA4 (250 mg/kg; oral suspension) group, and (5) the cisplatin (20 mg/kg) + GKA4 (250 mg/kg; oral suspension) + wortmannin (1.4 mg/kg; i.p.) group. The mice were orally administered with GKA4 for ten days. Control mice were orally administered saline daily. On day 7, AKI was induced in mice by an intraperitoneal injection of cisplatin (20 mg/kg) in the cisplatin and GKA4 groups. Wortmannin was given intraperitoneally to the animals of the intervention groups 1 h before cisplatin administration.

### 2.5. Kidney Index

Mice body weight was calculated before sacrifice. Then, kidneys were isolated and weighed, and the kidney index was measured as follows: kidney weight (mg)/body weight (g).

### 2.6. Renal Biomarker Measurements

The analysis of BUN and CRE levels to evaluate renal function was carried out using an ELISA reader (Roche Diagnostics, Cobas Mira Plus, Mannheim, Germany).

### 2.7. Histological Examination

Kidney tissue sections (5 µm) were processed with H&E and then photographed by light microscopy (Nikon, Eclipse, TS100, Tokyo, Japan). The degree of renal injury was classified into five grades (normal, <25% damage, 25–50% damage, 50–75% damage, and >75% damage) and was rated from 0 to 4 [17].

### 2.8. The TBARS (Thiobarbituric Acid Reactive Substance) Assay

The detection of TBARS was conducted by measuring the malondialdehyde (MDA) levels of renal lipid peroxidation [18]. Kidneys were lysed on ice with a lysis buffer. The thiobarbituric acid (TBA) solution was mixed with extracts at 90 °C for 45 min to form the MDA-TBA adduct. The amount of TBARS formed was measured at 532 nm against blank.

### 2.9. Measurement of Serum Cytokine Levels

Serum cytokine levels were detected using the BioLegend cytokine assay kit (BioLegend, San Diego, CA, USA).

### 2.10. Measurement of Serum NO Levels

Serum NO levels were measured using the Griess reaction colorimetric method [19]. An equal amount of Griess reagent was mixed with the culture supernatant, incubated at 540 nm for 10 min, and the absorbance was measured with a microplate reader (Molecular Devices, Orleans Drive, Sunnyvale, CA, USA).

### 2.11. Glutathione (GSH) Asaay

The assay was performed via the reaction of GSH and DTNB (5,5’-dithiobis (2-nitrobenzoic acid)). Briefly, 100 µL of supernatant, 200 µL of 0.3 M phosphate buffer (pH 8.4), 400 µL of double-distilled water, and 500 µL of Ellman’s reagent were mixed, and the absorbance was measured spectrophotometrically at 412 nm [20]. The total protein concentration was determined using a dye-binding method based on the Bradford assay (Bio-Rad Laboratories, Hemel Hempstead, UK).

### 2.12. Western Blot Analysis

Kidney tissue was homogenized with protease inhibitors in ice-cold RIPA buffer. For electrophoresis, antibodies were used as the primary antibodies. Anti-rabbit or anti-mouse IgG antibodies conjugated to HRP were used as the secondary antibodies. Membrane-bound HRP-labeled protein bands were monitored with chemiluminescent reagents, and signals were detected using Kodak Molecular Imaging Software (Eastman Kodak Company, Rochester, NY, USA).

### 2.13. Statistical Analysis

The experimental results are presented as the mean ± standard error of the mean (S.E.M). Following the assurance of the normal distribution of the data, we conducted a one-way ANOVA (ANOVA) with a Scheffé test, where *p* < 0.05 is the minimum requirement for statistical significance.

## 3. Results

### 3.1. GKA4 Inhibits Kidney Failure and Improves Kidney Activity after Cisplatin-Challenged AKI

The design of the experiment is shown in Figure 1A. BUN and CRE are two important tests of kidney function markers. As shown in Figure 1B, C, the oral administration of GKA4 (62.5, 125, and 250 mg/kg) inhibits the level of CRE and BUN after cisplatin-induced AKI. AMF is currently used in cancer treatment to reduce side effects and increase the survival of anticancer drugs. Thus, the results show that the pretreatment of GKA4 improves renal function in cisplatin-challenged AKI. Next, the analysis of renal tissue histopathology was used to determine whether GKA4 ameliorated cisplatin-induced AKI. As shown in Figure 1D, the control group had a normal renal tissue structure. Kidney tissue in the cisplatin-induced AKI was damaged by inflammatory cell infiltration, tubular epithelial cell loss, vacuolar degeneration, and necrosis. Thus, the oral administration of GKA4 reduced renal injury in mice with histological changes (Figure 1D,E). In addition, the oral administration of GKA4 significantly reduced renal failure scores compared with cisplatin-induced nephrotoxicity (Figure 1E). Thus, GKA4 ameliorates kidney function in cisplatin-treated AKI.

### 3.2. GKA4 Changes in the Renal Index against Cisplatin-Treated Mice

As shown in Table 1, cisplatin-treated mice had lighter body weights and had an increase in the relative renal index compared to the control mice. Additionally, it was found that the oral administration of GKA4 showed significantly higher resistance to cisplatin-induced nephrotoxicity and a reduced renal index.

### 3.3. GKA4 Reduces NO and Pro-Inflammatory Cytokine Levels in Cisplatin-Related Nephrotoxicity

Cisplatin-induced nephrotoxic effects appeared to increase in NO, TNF-α, IL-1β, and IL-6 secretion compared with data from the control group (Figure 2A–D). The oral administration of GKA4 suppressed NO, TNF-α, IL-1β, and IL-6 secretion following cisplatin induction. Thus, our report shows that the oral administration of GKA4 is associated with decreased levels of pro-inflammatory cytokines.

### 3.4. GKA4 Inhibits Oxidative Stress in Cisplatin-Related AKI

As shown in Figure 3A, B, cisplatin decreased the GSH levels in the antioxidant capacity and increased MDA levels in the lipid oxidation compared with data from the control group. Additionally, the oral administration of GKA4 decreased the levels of MDA and increased the GSH content. Therefore, GKA4 can diminish oxidative stress against cisplatin-treated AKI.

### 3.5. GKA4 Reduced the Inflammatory Response in Cisplatin-Induced AKI

Figure 4A shows the increased expression of iNOS and COX-2 protein in the cisplatin group compared with the control group and the reduced protein expression of iNOS and COX-2 in the renal tissues of the GKA4 group compared with data from the cisplatin-induced group.

Toll-like receptors (TLR) are immunosensors that recognize pathogen-associated molecular patterns and trigger signaling pathways associated with kidney injury. TLR-4 has an important regulatory position in renal pathology and can be a therapeutic target for alleviating renal injury. Figure 4B shows TLR-4 activation in cisplatin-treated AKI compared with the control group. Furthermore, GKA4 pretreatment significantly inhibited the protein expression of TLR-4 in cisplatin-associated nephrotoxicity. Thus, GKA4 regulated the TLR-4-mediated axis in cisplatin-induced AKI.

NF-κB activation is important for the inflammatory axis and has been implicated in various human diseases including kidney disease [21]. Figure 4B shows the phosphorylation of NF-κB and IκBα activation in renal tissues after cisplatin-induced AKI. In turn, GKA4 reduced the level of the phosphorylation of NF-κB and IκBα in the kidneys. Thus, the NF-κB axis is regulated by cisplatin-treated nephrotoxicity.

### 3.6. GKA4-Inactivated Cisplatin Induces the MAPK Pathway in Kidneys

MAPKs have important regulatory roles in cisplatin-induced kidney injury and inflammation [22]. The phosphorylation of MAPKs (p-JNK, p-ERK, and p-p38) in the renal tissues improved after cisplatin induction but were reduced by GKA4 (Figure 4C). Our report shows that the oral administration of GKA4 downregulated the phosphorylated MAPK protein expression against cisplatin-treated AKI.

### 3.7. GKA4 Restores Renal Antioxidant Defense and the HO-1/Nrf2 Signaling Pathway in Cisplatin-Associated Nephrotoxicity

Oxidative stress is considered to be one of the main causes of kidney damage [23]. As shown in Figure 5A, the protein expressions level of antioxidants (catalase, SOD1, and GPx3) diminished after cisplatin induction, but pretreatment with GKA4 restored this condition compared with the cisplatin group. In addition, the cisplatin group observed the reduced basal expression of Nrf2 and increased HO-1 expression compared with data from the control group (Figure 5B). Moreover, the oral administration of GKA4 can ameliorate the level of Nrf2 and HO-1 expression compared to the cisplatin group (Figure 5B). Novel findings showed that GKA4 is able to increase antioxidant enzyme-related protein expression after the cisplatin challenge.

### 3.8. GKA4 Alleviates the Cisplatin-Induced AMPK, SIRT1, PI3K, and AKT Expressions

Under oxidative stress conditions, p-CaMKK and p-AMPK can enhance the protein expression of SIRT1, thereby regulating energy balance and response [24]. As shown in Figure 6A, the levels of p-CaMKK, p-AMPK, and SIRT1 proteins were reduced after cisplatin induction. Furthermore, GKA4 obviously increased the p-CaMKK, p-AMPK, and SIRT1 protein expression after cisplatin induction. The results show that GKA4 increased the SIRT1, p-CaMKK, and p-AMPK protein expression after cisplatin exposure. In addition, the phosphorylation of AKT and PI3K was increased after cisplatin administration but decreased by GKA4 pretreatment. The results demonstrate that supplementation with GKA4 reduced the kidney damage by inhibiting PI3K/AKT signaling following a cisplatin challenge (Figure 6B).

### 3.9. GKA4 Decreases the Cisplatin-Induced Autophagy Signaling Pathway

The activation of autophagy is an important factor in alleviating cisplatin-induced AKI. The cisplatin group upregulated the protein expressions of LC3B, P62, and Beclin 1 compared with data from the control group. Otherwise, GKA4 pretreatment significantly suppressed the levels of LC3B, P62, and Beclin 1 after cisplatin exposure (Figure 7).

### 3.10. Blocking the Synergy of PI3K with Wortmannin Increases Renal Damage after Cisplatin Exposure

The design of the experiment is shown in Figure 8A. As shown in Table 2, the kidney index was increased in the cisplatin group compared with the control. Additionally, the GKA4 and/or wortmannin-treated group showed marked resistance to cisplatin-induced renal injury, such as a decreased kidney index and weight gain (Table 2). Moreover, only the wortmannin-treated group had higher CRE and BUN levels after the cisplatin challenge (Figure 8B, C). Conversely, the wortmannin and GKA4-treated group had lower CRE and BUN levels.

In addition, the analysis of histopathological changes was utilized to determine whether wortmannin and/or GKA4 had an effect on renal injury following the cisplatin challenge. The wortmannin-treated group demonstrated increased necrosis and inflammatory response against cisplatin-treated AKI. The wortmannin and GKA4-treated group showed reduced renal damage compared to the cisplatin group (Figure 8D,E).

### 3.11. GKA4 and/or Wortmannin Changes the Anti-Inflammatory Response

To evaluate whether wortmannin can suppress the PI3K/AKT axis, we assessed the related protein expression in the cisplatin + wortmannin group and cisplatin group. The results of both groups showed that the secretion of pro-inflammatory cytokines and MDA were increased, and GSH expression was decreased. Furthermore, the GKA4 + wortmannin group demonstrated that the expression of pro-inflammatory cytokines and the MDA level were reduced and the level of GSH was increased compared with the cisplatin group (Figure 9A–F). In addition, the GKA4 + cisplatin group could significantly ameliorate pro-inflammatory cytokines and MDA levels and an increased GSH level compared with the wortmannin + GKA4 + cisplatin group. Thus, the results demonstrated that GKA4 suppresses the activity of the PI3K/AKT axis after cisplatin-treated AKI.

### 3.12. GKA4 and Wortmannin Diminish the Inflammatory Secretions, Oxidative Stress, and PI3K/AKT Signal-Related Proteins

This experiment examined the level of protein expression of the PI3K/AKT axis associated with inflammatory and oxidative stress in cisplatin-treated mice treated with GKA4 or wortmannin to determine whether PI3K inhibitors could inhibit them. INEOS and COX-2 levels were improved after the cisplatin challenge (Figure 10A). In addition, co-treatment with GKA4 or wortmannin significantly reduced kidney tissues. The relative oxidative stress catalase, SOD1, and GPx3 levels were reduced after the cisplatin challenge. Moreover, co-treatment with GKA4 or wortmannin increased in kidney tissues (Figure 10B). The levels of the PI3K/AKT signal-related proteins p-pI3K and p-AKT were improved after the cisplatin challenge. Moreover, co-treatment with GKA4 or wortmannin reduced kidney tissue (Figure 10C). This indicates that GKA4 reduced the levels of the inflammatory cytokines, oxidative stress, and PI3K/AKT axis after cisplatin exposure.

## 4. Discussion

Cisplatin is a chemotherapeutic drug used to treat different types of cancers with a good prognosis, but the treatment still faces major challenges because of its numerous side effects, such as nausea, vomiting, nephrotoxicity, myelosuppression, and ototoxicity [2]. A known complication of cisplatin administration in patients is AKI. AKI has a mortality rate of up to 50%, and survivors of AKI have a substantially increased risk of developing chronic kidney disease [3]. Unfortunately, there is no effective way to prevent cisplatin-treated kidney injury. Therefore, there is an urgent need to develop new drugs for the treatment of AKI diseases.

Cisplatin is activated once it enters the cell. In the cytoplasm, the chlorine atoms on cisplatin are replaced by water molecules. This hydrolyzate is a potent electrophile that reacts with DNA and blocks cell division and leads to apoptosis [25]. Due to the increased solubility of cisplatin, the formation of cisplatin–GSH conjugates is inactivated, resulting in an increased rate of excretion from cells. Moreover, this process leads to depletion of intracellular GSH, which increases the toxicity of cisplatin [26].

The kinetics of cisplatin decay is biphasic in nature. Cisplatin concentration decreases in the kidney very rapidly after the initial accumulation of the drug (within 15 min) but then again increases and reaches the second peak 48–72 h after a single cisplatin administration [27]. About 43–50% of the cisplatin is eliminated in the urine in 24 h, 60–76% in 48 h, and about 91% in 72 h after single cisplatin administration (dose 4–10 mg/kg) [28]. At 72 h after cisplatin administration, the highest concentration of cisplatin was found in the mitochondria (37%), followed by cytosol (27%), nuclei (22%), and microsomes (14%) [29]. A cisplatin dose (LD_50_ = 6.6 mg/kg) significantly reduced the survival time of animals. For example, cisplatin at a dose of 20 mg/kg caused severe morphological damage in the kidneys and elevated BUN levels already 3 days after a single intraperitoneal injection, resulting in death within 5 days of injection [27]. Cisplatin at a dose of 40 mg/kg following a single cisplatin injection caused systemic toxicity within 1–2 days and death within 4 days [30]. Cellular uptake of cisplatin is mediated by transporters such as organic cation transporters (OCTs) and copper transporters (CTRs). These transporters are highly expressed in the proximal and distal tubules of the kidneys [31]. Inhibition of cisplatin influx transporters and activation of cisplatin efflux transporters have been popular strategies for reducing cisplatin-induced AKI. Thus, administration of cisplatin is shown to change the expression of cisplatin transporters. The toxicological effects of cisplatin are diverse and are an area of great concern for researchers.

The most commonly used animal model for studying cisplatin-induced AKI is a mouse model in which mice are administered one very high dose of cisplatin. The dose of cisplatin affects the degree of kidney damage. Current research on cisplatin-induced AKI mainly utilizes two mouse models: the short-term high-dose and long-term low-dose mouse models of nephrotoxicity. For example, long-term models use 5–15 mg/kg of cisplatin administered 2–4 times for 3–4 weeks. Short-term models use a single high dose of 20–30 mg/kg of cisplatin, which causes mortality and nephrotoxicity 3–7 days after cisplatin-induced AKI [1]. However, in the cisplatin rodent model, clinical signs develop with a delay of a few days and are progressive and dose related. They severely suffer from gastrointestinal malaise already 2 days after low nephrotoxic dose of cisplatin (6 mg/kg). At the dose of 17 mg/kg of cisplatin, clinical symptoms such as dehydration and decreased activity were evident on day 3, and serum BUN and CRE increased markedly on days 4–5. Death occurred 4–7 days after cisplatin administration [27].

Clinically, patients often receive low doses of cisplatin over an extended period of time to reduce the risk of nephrotoxicity while maintaining the drug’s therapeutic effect on cancer. The goal of cisplatin therapy is to kill the cancer, thereby increasing the lifespan of the patient [25,32]. Here, mice were administered GKA4 by oral gavage at doses of 62.5, 125, and 250 mg/kg once daily for 10 days (7 days before and 3 days after cisplatin injection). Control mice were orally administered saline daily. On day 7, AKI was induced in mice by intraperitoneal injection of cisplatin (20 mg/kg) in the cisplatin and GKA4 groups.

Lactic acid bacteria (LAB) are often used for nutritional value, flavor, and aroma enhancers in foods. Moreover, these properties are produced by these bacteria through fermentation to produce various sugars and metabolites, such as lactic acid, acetic acid, ethanol, diacetyl, acetone, exopolysaccharides, and bacteriocins [33]. LABs of the genera Lactobacillus, Bifidobacterium, Saccharomyces, Streptococcus, Pediococcus, Leuconostoc, and Bacillus are found in human gut microbiota and are classified as probiotics. *P. acidilactici* is extremely resistant to the acidic environment of the stomach. The survival rate of *P. acidilactici* in the stomach is greater than most other probiotic strains, allowing it to more easily populate the small intestine and provide benefits to the gut. In the small intestine, *P. acidilactici* can help improve health by producing lactic acid and a variety of antimicrobial compounds that diminish the harmful pathogens in the gut [34].

*P. acidilactici* is capable of producing phenolic compounds that have demonstrated inhibitory properties against molds and fungi in many foods. Furthermore, research has shown the therapeutic possibilities of *P. acidilactici* for the treatment of multiple sclerosis and the ability to prevent autoimmune diseases and reduce constipation, diarrhea, and stress [35,36]. Many studies on *P. acidilactici* are currently underway, and there is evidence that it may provide a wide range of beneficial effects [37]. There may even be more benefits that have yet to be discovered.

The pathway of cisplatin-treated AKI is still unclear. However, some evidence suggests that oxidative damage and inflammatory reaction lead to kidney damage [38]. In this experiment, the use of the intraperitoneal administration of cisplatin to induce AKI in mice is a new animal model [39]. We applied this mouse model to confirm the defense of GKA4 against oxidative and inflammatory stress and to examine whether GKA4 is protective against cisplatin-challenged AKI. In addition, AMF (positive control) acts by phosphorylating alkaline phosphatase to its active metabolite, which protects normal cells from chemotherapy and radiation therapy and enhances the therapeutic effect. However, AMF does not change the antitumor effect but prevents cisplatin damage [39]. In our study, oral GKA4 was found to prevent cisplatin-induced AKI because it is a natural food with few side effects.

Nephrotoxicity is a problem for cisplatin as a treatment for various tumors. Cisplatin can accumulate in renal tubular cells after injection, leading to the loss of kidney function. The oral demonstration of GKA4 diminished the levels of BUN and CRE on day 10 after cisplatin-challenged AKI, suggesting that GKA4 raises renal activity. It has been studied that *P. acidilactici* can survive through the gastrointestinal tract, can survive drying, can be stored at 4 °C for 60 days [40,41], and is active against *Helicobacter pylori* in vitro [42]. It has been shown that *P. acidilactici* ameliorates liver fibrosis induced by carbon tetrachloride (CCl_4_) [43].

Cisplatin has been shown to cause severe kidney damage such as acute tubular necrosis and tubular cell damage [3]. In this study, severe tubular necrosis was also found in cisplatin-induced mice. Due to the protective effect of GKA4, there was almost no tubular necrosis in the GKA4-administered group, demonstrating the renoprotective effect of GKA4 against cisplatin-related AKI. Numerous studies have recommended that oxidative stress, inflammation, and renal vascular damage may play a critical role in the pathogenesis of cisplatin-treated AKI [27]. Cisplatin can accumulate in proximal tubules and cause renal cytotoxicity, and damaged renal tubular epithelial cells can induce an inflammatory response [44]. The present study demonstrated that GKA4 reduces the levels of pro-inflammatory cytokines following the cisplatin challenge, which is important for the mechanism by which GKA4 ameliorates cisplatin-related AKI.

NF-kB plays an important role in inflammation by controlling pro-inflammatory factors. Cisplatin-induced stress activates the phosphorylation of IκB proteins, which subsequently degrade and release NF-κB. NF-κB activation translocates into the nucleus and leads to the control of genes for the expression of COX-2, iNOS, and pro-inflammatory cytokines in the renal tissues [45]. In this study, the regulation of the protein expression levels of COX-2, iNOS, p-IκB, and p-NF-κB proteins was considerably improved in cisplatin-related AKI. However, the expression of these proteins was found to be inhibited after the oral administration of GKA4. Thus, GKA4 can block cisplatin-induced AKI by inhibiting the inflammatory axis.

TLR-4 is a receptor protein that upregulates the NF-κB pathway and the production of pro-inflammatory cytokines. The activation of TLR-4 could result in severe inflammation and kidney damage [46]. Cisplatin induction increases the expression of TLR4 in cells, which is important in cisplatin-induced nephrotoxicity that inhibits the production of inflammatory mediators in the kidneys. Studies have shown that cisplatin and TLR4-specific ligand lipopolysaccharide (LPS) synergistically produce a large number of cytokines and, thus, lead to nephrotoxicity [46]. Cisplatin-treated inflammatory response and kidney damage were significantly diminished in TLR4-null mice compared to the wild-type mice [47]. Our experimental results show that the levels of TLR-4, p-IκB, and p-NF-κB are increased after cisplatin treatment but can be reversed by GKA4. Other studies have also shown that the activation of NF-κB is associated with cisplatin-challenged AKI in patients and animal models after the cisplatin challenge.

MAPKs play important roles in regulating cisplatin-induced renal injury and inflammatory response [48]. In the present study, the oral administration of GKA4 significantly inhibited the phosphorylation of MAPKs, so much so that GKA4 can diminish the occurrence of acute renal failure after cisplatin administration. GKA4 also avoids raising the level of pro-inflammatory cytokines by activating NF-κB in cisplatin-induced AKI. In conclusion, GKA4 could significantly avoid the phosphorylation of MAPK and NF-κB and the degradation of IκB-α after cisplatin exposure.

Oxidative stress is an important point after cisplatin-treated AKI [49]. Oxidative stress can also damage cells by increasing the lipid peroxidation and reducing the expression of antioxidant enzymes [50]. GKA4 increased the SOD, catalase, and GPx expressions, as well as the GSH levels, and decreased the MDA formation after cisplatin induction. In addition, oxidative stress leads to the dissociation of Nrf2 to activate downstream gene expression related to oxidative stress and inflammation such as HO-1 and GPx, which protect kidney tissues by eliminating oxidative damage [51]. Our experimental results suggest that the protective role of GKA4 modulates the Nrf2/HO-1 axis after cisplatin induction. However, GKA4 pretreatment can effectively reverse the changes of oxidative stress.

The metabolism of lactic acid bacteria and its antioxidant activity can be attributed to its antioxidant enzymes, and the production of GSH, which upregulate the antioxidant activity of the host and increase the level of antioxidant metabolites in the body [52]. This affects the host, the regulation of signaling pathways, the downregulation of ROS-producing enzymes, and the modulation of gut microbiota. The chemopreventive properties of LAB have been demonstrated in numerous in vitro, in vivo, and human clinical studies and are mediated through multiple mechanisms, including alterations in the gastrointestinal microbiota, increased host immune responses, and antioxidant and antiproliferative effects [53]. According to research, the antioxidant activity of LAB is due to their metal chelating ability and ROS scavenging ability. Therefore, it has a major impact on diseases associated with imbalances in the gut microbiome such as inflammatory disease, diabetes, and cancer [54].

Autophagy elevated cell viability by eliminating damaged organelles by facilitating cellular bioenergetic homeostasis. Currently, autophagy is considered to be an inducible and regulated mechanism that closely evaluates cell survival or death during renal disease. Furthermore, autophagy is also a defense mechanism against nutrient deficiencies, aging, and pathogen invasion in inflammation-related diseases [55]. For example, during nutrient starvation, autophagy is used to eliminate damaged organelles, generate ATP, and promote protein synthesis. The excessive activation of oxidative stress and inflammation can cause damage to the body, and the emergence of autophagy can affect the expression levels of autophagy-related proteins LC3-II/I, Beclin 1, and p62 [56]. Our findings show that GKA4 decreased the protein expression of LC3-II, p62, and Beclin 1, indicating that autophagy is associated with cisplatin-induced AKI activation in mice.

AMPK is a metabolic regulator that restores energy balance during metabolic stress. Interestingly, the levels and activities of AMPK and SIRT1 in the kidney are significantly reduced when the kidney is damaged [55]. These findings were also confirmed in a cisplatin-induced AKI. The AMPK/SIRT1 pathway plays an integral role in assisting cell survival and mitochondrial biosynthesis, inhibiting inflammation and apoptosis and stimulating antioxidant synthesis [56]. SIRT1 modulated its downstream pathways by targeting several cellular proteins such as NF-κB, thereby avoiding ROS at the cellular level [57]. In this study, we found that NF-κB can be activated by SIRT1 and that the oral administration of the GKA4 treatment inhibited the activation of NF-κB to reduce oxidative stress-induced renal tissue damage and inflammation.

AMPK, SIRT1, and the NF-κB axis have important roles in renal damage and oxidative stress [58,59], so we orally administered GKA4 to inhibit cisplatin-induced AKI via the AMPK/SIRT1/NF-κB signaling progress. The relative expression of p-AMPK and p-SIRT1 was reduced in the cisplatin-treated damage, but the oral administration of GKA4 reversed these changes. Therefore, GKA4 acts primarily on the activation of AMPK, thereby promoting its downstream target molecule SIRT1 and subsequently inhibiting the action of NF-κB. Taken together, cisplatin-related AKI diminished the expression level of SIRT1, p-AMPK, and p-NF-κB, and their expression may be increased after treatment with GKA4.

PI3K and AKT control cellular processes by phosphorylating substrates involved in the regulation of growth, proliferation, survival, motility, metabolism, and immune response [60]. Importantly, the PI3K/AKT signaling pathway can be activated by insulin-like growth factor 1 (IGF-1) to regulate renal repair after ischemia/reperfusion (I/R) injury [61,62]. The relative expression of p-PI3K and p-AKT was significantly increased in the cisplatin-induced AKI model, but the oral administration of GKA4 reversed these changes. Therefore, from these data, it seems very logical that GKA4 acts primarily on the inhibition of PI3K/AKT signaling. Taken together, our data suggest that cisplatin-related AKI increased the expression of p-PI3K and p-AKT, and that their expression may be increased after treatment with GKA4. SIRT1 has been found to mediate several inflammation-related gene expressions, such as NF-кB. Thus, the data suggest that specific PI3K inhibitors increase the activation of inflammatory pathways, and GKA4 effectively reverses these inflammatory changes after cisplatin-treated AKI.

## 5. Conclusions

In this article, we demonstrated that GKA4 regulates inflammatory effects and oxidative stress after cisplatin-treated AKI by the repression of kidney histopathologic changes, inflammatory cell infiltration, and the release of pro-inflammatory cytokines. GKA4 can exhibit potent anti-inflammatory and antioxidant effects against cisplatin-associated AKI, mediated by inhibiting the TLR-4/NF-κB/MAPK, HO-1/Nrf2, AMPK/SIRT1/NF-κB and PI3K/AKT axes (Figure 11). These current studies suggest that GKA4 is renoprotective only in a mouse model of cisplatin nephrotoxicity. More research is needed to confirm its effectiveness. In conclusion, GKA4 inhibits inflammatory response and oxidative stress in cisplatin-treated AKI to prevent renal injury.

## Figures and Tables

**Figure 1 nutrients-14-02877-f001:**
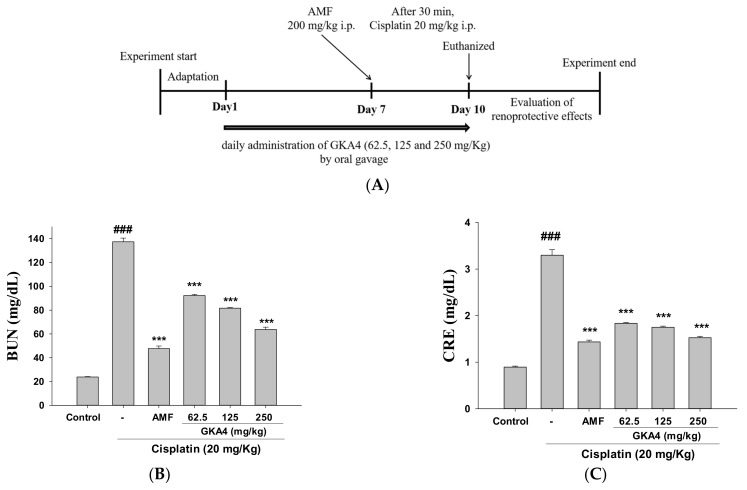
The design of the experiment (**A**) and the renoprotective effects of GKA4 against cisplatin-treated neurotoxicity. Mice were administered orally with GKA4 at doses of 62.5, 125, and 250 mg/kg every day for 10 days, with GKA4 first on day 7 and cisplatin 1 h later, and mice were sacrificed on day 11. The figure shows serum BUN levels (**B**), CRE levels (**C**), kidneys stained with H&E (**D**), and the kidney injury scores (**E**). Histopathologic analysis of kidney tissues using H&E staining and photographed (400×). The data are presented as the means ± S.E.M (*n* = 5). ### denotes *p* < 0.001 compared with the sample of the control group. *** *p* < 0.001 compared with the cisplatin group. GKA4: P. acidilactici GKA4; AMF: amifostine, BUN: blood urea nitrogen; CRE: creatinine.

**Figure 2 nutrients-14-02877-f002:**
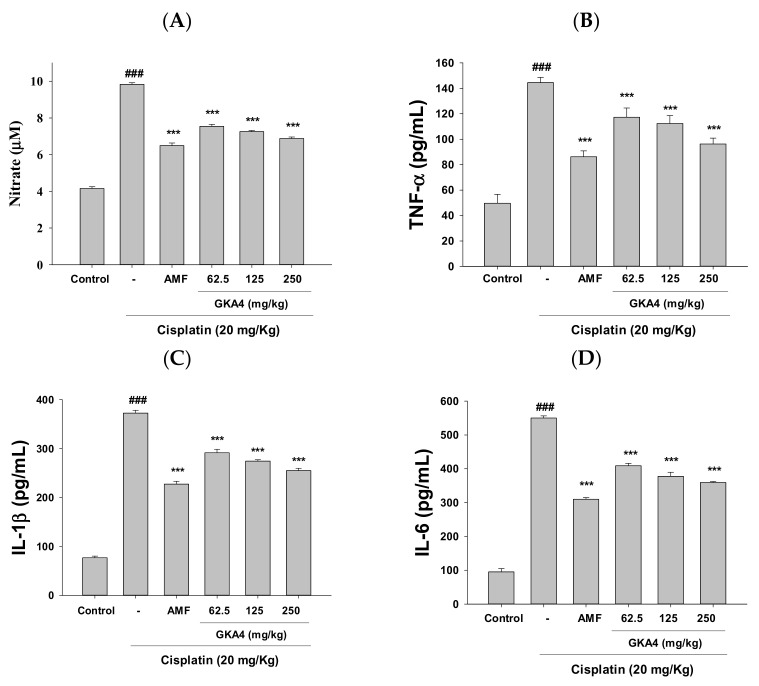
GKA4 suppressed (**A**) NO, (**B**) TNF-α, (**C**) IL-1β, and (**D**) IL-6 levels in cisplatin-treated AKI. The Griess reaction assay determined the nitrite concentration. ELISA kits were used to evaluate the levels of pro-inflammatory cytokines. The data are presented as the means ± S.E.M (*n* = 5). ^###^ denotes *p* < 0.001 compared with the sample of the control group. *** *p* < 0.001 compared with the cisplatin group.

**Figure 3 nutrients-14-02877-f003:**
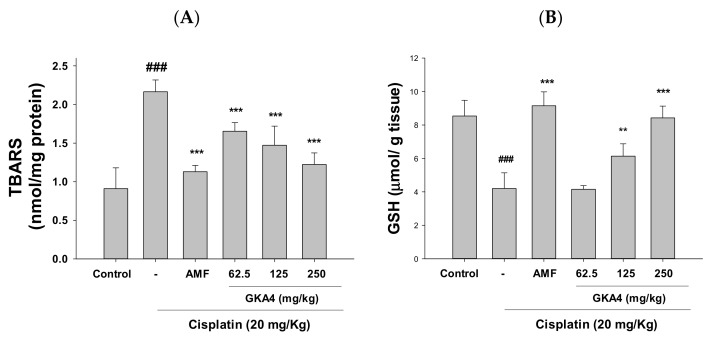
GKA4 avoids oxidative stress against cisplatin-treated AKI. MDA (**A**) and GSH levels (**B**) were measured by MDA and GSH assays. The values were displayed as means ± S.E.M (*n* = 5). ### denotes *p* < 0.001 compared with the sample of the control group. ** *p* < 0.01 and *** *p* < 0.001 compared with the cisplatin group. GSH: glutathione.

**Figure 4 nutrients-14-02877-f004:**
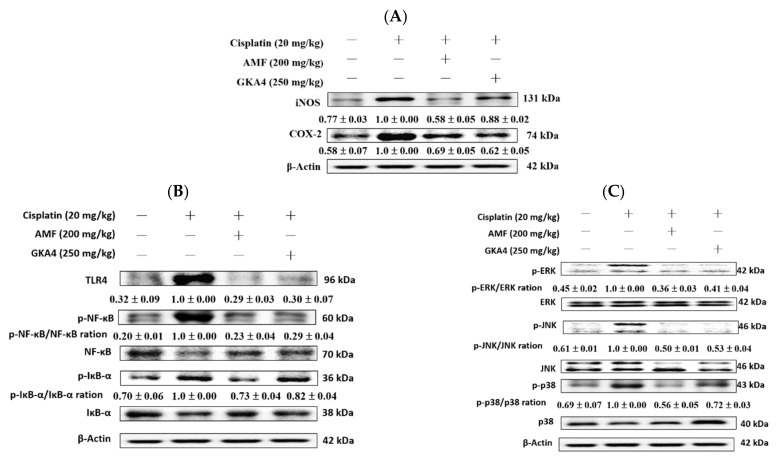
GKA4 inhibited the cisplatin-treated (**A**) iNOS and COX-2, (**B**) TLR-4, IκB-α, p-IκB-α, p-NF-κB, and NF-κB, and (**C**) MAPK phosphorylation protein levels against cisplatin-induced nephrotoxicity. The protein levels of iNOS, COX-2, TLR-4, IκB-α, NF-κB, and MAPK phosphorylation protein expression in renal tissues were analyzed by western blot after cisplatin induction. Protein bands were analyzed by densitometric analysis. Experiments were performed at least three times, and a representative one is shown. Plus signs: with drug; Minus signs: without drug.

**Figure 5 nutrients-14-02877-f005:**
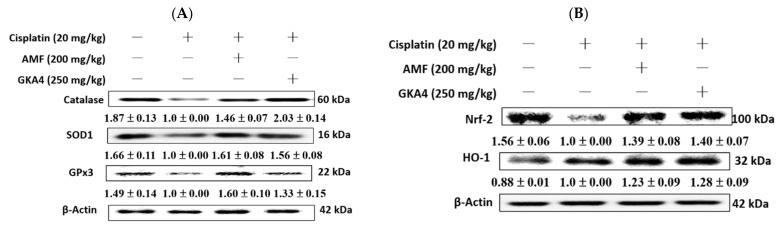
In renal tissues, the effects of GKA4 on the protein expression induced by cisplatin, including (**A**) anti-oxidative enzymes (catalase, SOD1, and GPx3) and (**B**) HO-1 and Nrf2. The protein levels of HO-1 and Nrf2 protein expression in renal homogenates were assessed by western blot analysis after the cisplatin challenge. Protein bands were analyzed by densitometric analysis. Experiments were performed at least three times, and representative images are shown. Plus signs: with drug; Minus signs: without drug.

**Figure 6 nutrients-14-02877-f006:**
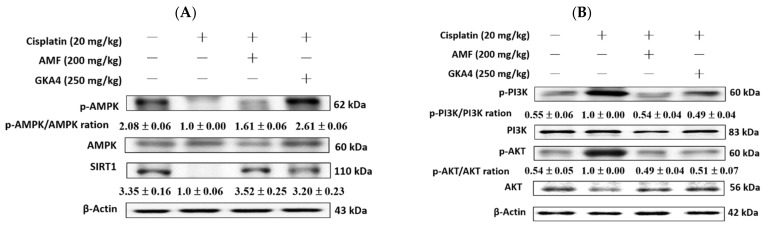
GKA4 regulated AMPK, SIRT1, PI3K, and AKT protein expression in cisplatin-induced AKI mice. Kidney tissue lysate was subjected to western blot analysis using antibodies specific to AMPK, p-AMPK SIRT1, PI3K, p-PI3K, AKT, and p-AKT. GKA4 regulated AMPK/SIRT1 signal (**A**) and PI3K/AKT signal pathway (**B**). Protein bands were analyzed by densitometric analysis. Experiments were performed at least three times, and representative images are shown. Plus signs: with drug; Minus signs: without drug.

**Figure 7 nutrients-14-02877-f007:**
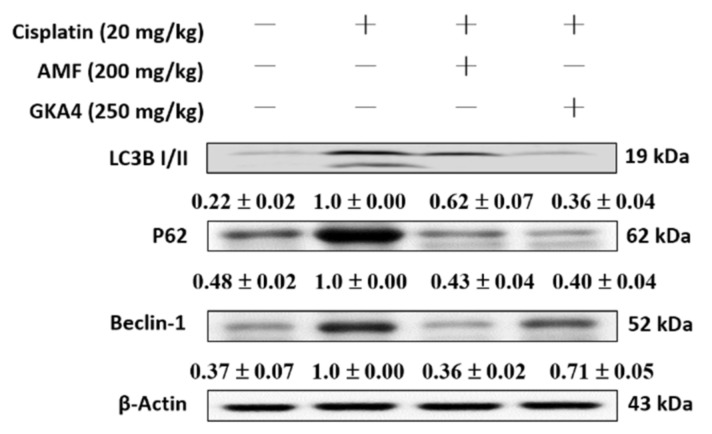
GKA4 diminished the expression of LC3B, P62, and Beclin 1 proteins after cisplatin exposure. Kidney tissue lysate was subjected to western blot analysis using antibodies specific to LC3B, P62, Beclin 1, and β-actin. Protein bands were analyzed by densitometric analysis. Experiments were repeated at least three times, and representative images are shown. Plus signs: with drug; Minus signs: without drug.

**Figure 8 nutrients-14-02877-f008:**
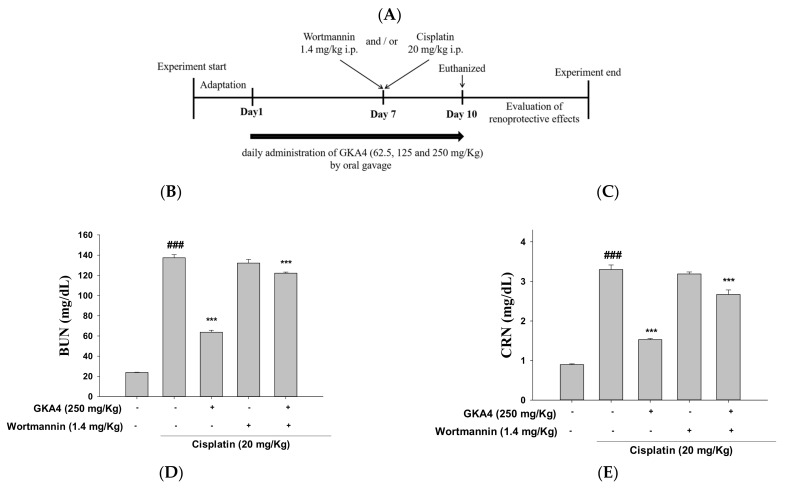
Effects of GKA4 and the PI3K inhibitor (wortmannin) after cisplatin exposure. GKA4 (250 mg/kg) and/or wortmannin (1.4 mg/kg) were given daily for 10 days. Cisplatin (20 mg/kg, i.p.) was administered 1 h after GKA4 and/or wortmannin on day 7, and euthanasia was performed on day 11. The figure shows the design of the experiment (**A**), BUN levels (**B**), CRE levels (**C**), kidneys stained with H&E (**D**), and kidney injury scores (**E**). Histopathologic analysis of kidney tissues using H&E staining and photographed (400×). The values were displayed as the means ± S.E.M (*n* = 5). The data are presented as the means ± S.E.M (*n* = 5). ### denotes *p* < 0.001 compared with the sample of the control group. *** *p* < 0.001 compared with the cisplatin group.

**Figure 9 nutrients-14-02877-f009:**
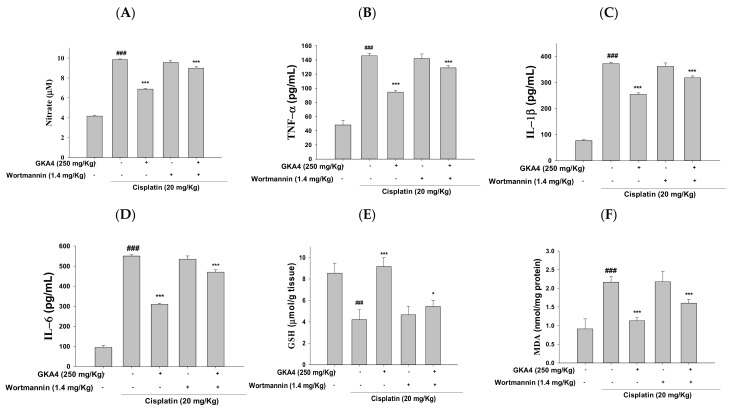
GKA4 and wortmannin changed the NO (**A**), TNF-α (**B**), IL-1β (**C**), IL-6 (**D**), GSH (**E**), and MDA (**F**) levels after cisplatin-treated AKI. The Griess reaction assay determined the nitrite concentration. ELISA kits were used to evaluate the levels of pro-inflammatory cytokines. The values were displayed as means ± S.E.M (*n* = 5). ^###^ denotes *p* < 0.001 compared with the sample of the control group. * *p* < 0.05 and *** *p* < 0.001 compared with the cisplatin group.

**Figure 10 nutrients-14-02877-f010:**
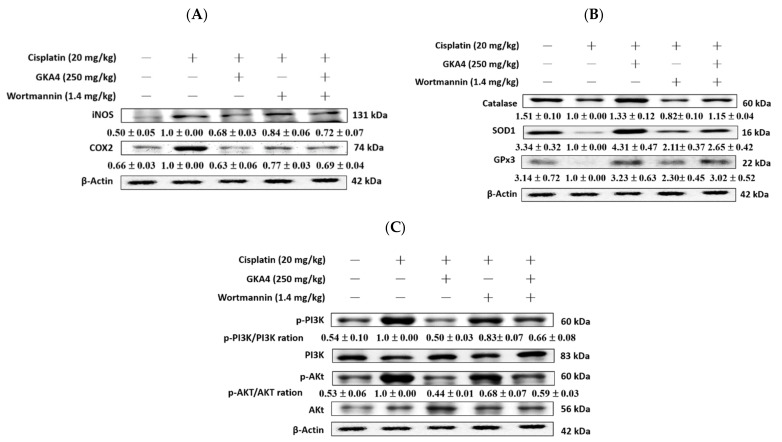
GKA4 and wortmannin-regulated inflammatory (**A**), oxidative stress (**B**), and PI3K/AKT signal-related proteins (**C**) after cisplatin-challenged AKI. Kidney tissue lysate was subjected to western blot analysis using specific antibodies for iNOS, COX-2, catalase, SOD1, GPx3, p-PI3K, PI3K, p-AKT, and AKT. Protein bands were analyzed by densitometric analysis. Experiments were repeated at least three times, and the representative images are shown. Plus signs: with drug; Minus signs: without drug.

**Figure 11 nutrients-14-02877-f011:**
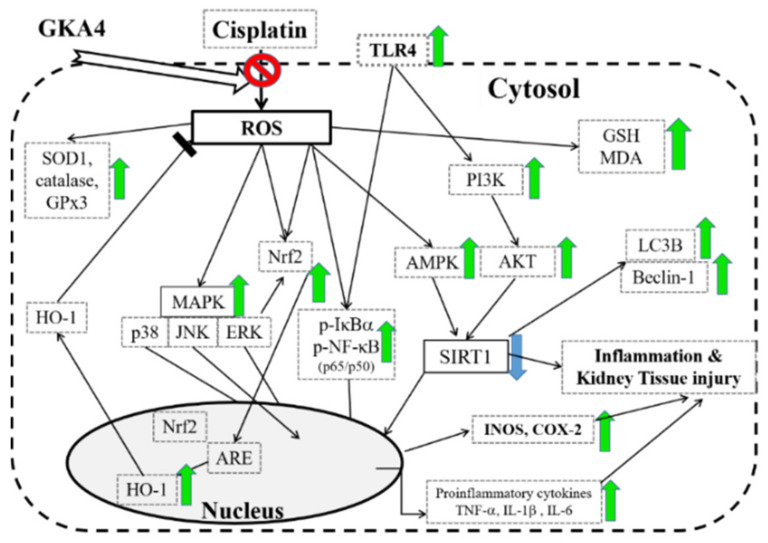
A scheme displaying the protective effect of GKA4 against cisplatin-associated renal injury. The green arrows indicate an increase. The blue arrows indicate a decrease. ROS: reactive oxygen species; MDA: malondialdehyde; GSH: glutathione; MAPK: mitogen-activated protein kinase; JNK: C-jun NH2-terminal kinase; ERK: extracellular-signal-regulated kinase; ARE: antioxidant response element; AP-1: activator protein 1; NF-κB: nuclear factor of κB; HO-1: heme oxygenase 1; SOD1: Cu/Zn superoxide dismutase; GPx3: glutathione peroxidases 3; Nrf2: nuclear-factor-erythroid-2-related factor 2; TLR-4: toll-like receptor 4; IκB: inhibitor of the nuclear factor kappa B; AMPK: 5’-adenosine-monophosphate-activated protein kinase; SIRT1: sirtuin-1; iNOS: inducible nitric oxide synthase; COX-2: cyclooxygenase-2; LC3B: microtubule-associated protein1 light chain 3; PI3K: phosphatidylinositol-3 kinase; AKT: protein kinase B (PKB); TNF-α: tumor necrosis factor-α; IL-1ß: interleukin-1β; IL-6: interleukin-6.
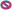
: inhibition;

: up-regulation;

: down-regulation.

**Table 1 nutrients-14-02877-t001:** Oral administration of GKA4 alters the body weight and renal index in cisplatin-related nephrotoxicity. The values were displayed as means ± SEM (*n* = 5).

Groups	Initial Body Weight (g)	Final Body Weight (g)	Kidney Index (mg/g)
Control	34.12 ± 0.82	39.52 ± 0.47	1.36 ± 0.08
Cisplatin (20 mg/kg)	34.14 ± 0.15	32.50 ± 0.16 ^###^	2.36 ± 0.09 ^###^
Cisplatin (20 mg/kg) + AMF (200 mg/kg)	34.08 ± 0.30	36.56 ± 0.15 ***	1.51 ± 0.08 ***
Cisplatin (20 mg/kg) + GKA4 (62.5 mg/kg)	34.06 ± 0.44	35.48 ± 0.22 ***	1.85 ± 0.09 ***
Cisplatin (20 mg/kg) + GKA4 (125 mg/kg)	34.14 ± 0.35	35.84 ± 0.15 ***	1.75 ± 0.04 ***
Cisplatin (20 mg/kg) + GKA4 (250 mg/kg)	34.06 ± 0.21	36.36 ± 0.21 ***	1.61 ± 0.06 ***

### denotes *p* < 0.001 compared with the sample of the control group. *** *p* < 0.001 compared with the cisplatin group. The kidney index is the ratio of the kidney weight to body weight.

**Table 2 nutrients-14-02877-t002:** GKA4 and the PI3K inhibitor (wortmannin) change the body weight and the kidney index, showing resistance to cisplatin-associated nephrotoxicity. The values were displayed as means ± S.E.M (*n* = 5).

Groups	Initial Body (g)	Final Body (g)	Kidney Index (mg/g)
Control	34.00 ± 0.69	39.56 ± 0.55	1.36 ± 0.08
Cisplatin (20 mg/kg)	33.90 ± 0.16	36.50 ±0.19 ^###^	2.36 ± 0.09 ^###^
Cisplatin (20 mg/kg) + GKA4 (250 mg/kg)	34.10 ± 0.13	34.30 ± 0.53 ***	1.61 ± 0.04 ***
Cisplatin (20 mg/kg) + wortmannin (1.4 mg/kg)	34.40 ± 0.26	33.40 ± 0.51 *	2.26 ± 0.08
Cisplatin (20 mg/kg) + GKA4 (250 mg/kg) + wortmannin (1.4 mg/kg)	33.88 ± 0.26	34.32 ± 0.19 ***	2.05 ± 0.06 ***

### denotes *p* < 0.001 compared with the sample of the control group. * *p* < 0.05 and *** *p* < 0.001 compared with the cisplatin group.

## Data Availability

Not applicable.

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
