# Peer review of "Renoprotective Effect of Pediococcus acidilactici GKA4 on Cisplatin-Induced Acute Kidney Injury by Mitigating Inflammation and Oxidative Stress and Regulating the MAPK, AMPK/SIRT1/NF-κB, and PI3K/AKT Pathways"

_nutrients, 2022, doi:10.3390/nu14142877_

Round 1
Reviewer 1 Report
I have already been a reviewer of this manuscript in the past and unfortunately I continue to point out the same problem, which the authors do not intend to solve, but which for me is absolutely to be addressed. Due to the way in which the experiments were conducted, it cannot be excluded that the observed effect depends on a decreased bioavailability of cis-platinum due to the co-administration with the probiotic, rather than on an effect of the latter on the toxicity of the drug. In order for the manuscript to be suitable for publication, it is therefore necessary to either measure the drug levels in the different conditions, or think of a different treatment protocol
Author Response
Dear Editor-in-chief,
We appreciated very much for your kind consideration as regards revising the manuscript entitled "Renoprotective effect of Pediococcus acidilactici GKA4 on cisplatin-induced acute kidney injury by mitigating Inflammation and Oxidative Stress and Regulating MAPK, AMPK/SIRT1/NF-κB, and PI3K/AKT Pathway” (Manuscript ID: nutrients-1781904R1). This manuscript has been re-evaluated according to the comments given by your panel of reviewers. We made a point-by-point response to each comment indicated by the reviewers, which are all contained in the attached “Response to the Reviewers”. We also have highlighted the changes in the text with red color for the ease of your review.
Lastly, we would like to thank you once again for providing us the opportunity to improve our manuscript.
Lastly, we would like to thank you once again for providing us the opportunity to improve our manuscript.
Sincerely yours,
Guan-Jhong Huang, Ph. D
School of Chinese Pharmaceutical Sciences and Chinese Medicine Resources,
College of Chinese Medicine, China Medical University, Taichung 404, Taiwan
E-mail: [email protected]
Point-by-point Response to the Reviewers
Reviewers' 1 comments:
Comments and Suggestions for Authors
I have already been a reviewer of this manuscript in the past and unfortunately I continue to point out the same problem, which the authors do not intend to solve, but which for me is absolutely to be addressed. Due to the way in which the experiments were conducted, it cannot be excluded that the observed effect depends on a decreased bioavailability of cis-platinum due to the co-administration with the probiotic, rather than on an effect of the latter on the toxicity of the drug. In order for the manuscript to be suitable for publication, it is therefore necessary to either measure the drug levels in the different conditions, or think of a different treatment protocol
[Reply]: Thank you for your advice. The reviewer wants to measure the drug levels in the different conditions, or think of a different treatment protocol. At present, in all acute kidney model we did not find to measure the drug levels. Our laboratory does not do experiments related to pharmacokinetics, but only studies some nutrients that are beneficial to the human body and related researches on their mechanisms. And the reviewers want to see different treatment options, that is not the induction of acute kidney injury caused by cisplatin model. Your suggestion is good, but not a short-term solution at present. The current study found that when renal tubular cells are exposed to cisplatin, a cascade of reactions is initiated that ultimately leads to renal tubular cell damage and even death. The inflammatory response caused by cisplatin also aggravates the damage of renal tubular cells [1].
In addition, there are only two published studies on the relationship between probiotics and acute kidney injury. Publishing findings in Nutrients 2021 that cisplatin-induced nephrotoxicity is associated with gut microbiota disturbance. Supplementation of Lactobacillus reuteri and Clostridium butyricum (LCs) had a protective effect on cisplatin-induced nephrotoxicity through reconstruction of gut microbiota. LCs exhibited significantly decreased renal inflammation (KIM-1, F4/80, and MPO), oxidative stress, fibrosis (collagen IV, fibronectin, and a-SMA), apoptosis, concentration of blood endotoxin and indoxyl sulfate, and increased fecal butyric acid production compared with those without supplementation [2]. Another publishing finding in IJMS 2021 that Lactobacillus salivarius BP121 prevented cisplatin‑induced AKI in rats, as demonstrated by decreases in inflammation and oxidative stress in kidney tissue and in serum levels of uremic toxins such as indoxyl sulfate (IS) and p‑cresol sulfate (PCS). BP121 also reduced intestinal permeability, as determined using fluorescein isothiocyanate‑dextran by immunohistochemical detection of tight junction (TJ) proteins such as zona occludens‑1 and occludin. The abundance of Lactobacillus spp., which are beneficial intestinal flora, was increased by BP121; this was accompanied by an increase in the concentrations of short‑chain fatty acids in feces [3].
Reference:
- Holditch SJ, Brown CN, Lombardi AM, Nguyen KN, Edelstein CL. Recent Advances in Models, Mechanisms, Biomarkers, and Interventions in Cisplatin-Induced Acute Kidney Injury. Int J Mol Sci. 2019, 20(12), 3011.
- Hsiao YP, Chen HL, Tsai JN, Lin MY, Liao JW, Wei MS, Ko JL, Ou CC. Administration of Lactobacillus reuteri Combined with Clostridium butyricum Attenuates Cisplatin-Induced Renal Damage by Gut Microbiota Reconstitution, Increasing Butyric Acid Production, and Suppressing Renal Inflammation. Nutrients. 2021, 13(8):2792.
- Lee TH, Park D, Kim YJ, Lee I, Kim S, Oh CT, Kim JY, Yang J, Jo SK. Lactobacillus salivarius BP121 prevents cisplatin‑induced acute kidney injury by inhibition of uremic toxins such as indoxyl sulfate and p‑cresol sulfate via alleviating dysbiosis. Int J Mol Med. 2020, 45(4):1130-1140.
Reviewer 2 Report
The authors have made significant improvements in the manuscript and have answered my comments in a scientific way with reference,.
Author Response
Dear Editor-in-chief,
We appreciated very much for your kind consideration as regards revising the manuscript entitled "Renoprotective effect of Pediococcus acidilactici GKA4 on cisplatin-induced acute kidney injury by mitigating Inflammation and Oxidative Stress and Regulating MAPK, AMPK/SIRT1/NF-κB, and PI3K/AKT Pathway” (Manuscript ID: nutrients-1781904R1). This manuscript has been re-evaluated according to the comments given by your panel of reviewers. We made a point-by-point response to each comment indicated by the reviewers, which are all contained in the attached “Response to the Reviewers”. We also have highlighted the changes in the text with red color for the ease of your review.
Lastly, we would like to thank you once again for providing us the opportunity to improve our manuscript.
Sincerely yours,
Guan-Jhong Huang, Ph. D
School of Chinese Pharmaceutical Sciences and Chinese Medicine Resources,
College of Chinese Medicine, China Medical University, Taichung 404, Taiwan
E-mail: [email protected]
Point-by-point Response to the Reviewers
Reviewers' 2 comments:
Comments and Suggestions for Authors
The authors have made significant improvements in the manuscript and have answered my comments in a scientific way with reference.
[Reply]: Thank you

This manuscript is a resubmission of an earlier submission. The following is a list of the peer review reports and author responses from that submission.
Round 1
Reviewer 1 Report
The manuscript “Renoprotective effect of Pediococcus acidilactici GKA4 on cisplatin-induced acute kidney injury by mitigating Inflammation and Oxidative Stress and Regulating MAPK, AMPK/SIRT1/NF-κB, and PI3K/AKT Pathway” by Wen-Hsin Lin et al. They have; reported that Pediococcus acidilactici GKA4 could play an important role in the Nephroprotection against cisplatin. The manuscript is written in good English language with few grammatical errors. Based on the current status of this manuscript, I think a major revision is necessary.
Comments:
- The authors should check grammatical errors very carefully before resubmission.
- In the introduction lines, no 1-11 have not been cited properly, I will suggest the authors cite these lines with recent references and add about P. acidilactici Grape King A4 strain (GKA4) with recent references in line no 41-46
- Section 2.4. The research Design is confusing…. like on day 7, AKI was induced in mice by an intraperitoneal injection of cisplatin (20 mg/kg) in the cisplatin and GKA4 groups. The mice were sacrificed on day ten after cisplatin injection…it is confusing? Did the authors perform GKA4 and wortmannin at two different times? If not then why control and cisplatin groups have been repeated?
- In section 2.12. In western Blot Analysis, authors should add the specific primary and secondary antibodies along with their dilutions.
- On page 13 line no 11 “AKI has a mortality rate of up to 50%” authors have mentioned there is 50% due to AKI, then why only 5 animals were taken in each group? Did the authors notice any mortality? Should be added to the manuscript.
- Representation of statistical notation (a..b….c.d ) is very difficult to understand. Explain statically significance in a simple but explanatory and easily understandable way.
- Did the authors perform only a single set of protein expressions by western blot? Without statics. Results can not be trusted, so, multiple sets of blots must be performed, however, only a single set can be used in the manuscript but, an average of all sets must be mentioned.
Author Response
Dear Editor-in-chief,
We appreciated very much for your kind consideration as regards revising the manuscript entitled "Renoprotective effect of Pediococcus acidilactici GKA4 on cisplatin-induced acute kidney injury by mitigating Inflammation and Oxidative Stress and Regulating MAPK, AMPK/SIRT1/NF-κB, and PI3K/AKT Pathway” (Manuscript ID: nutrients-1726894R1). This manuscript has been re-evaluated according to the comments given by your panel of reviewers. We made a point-by-point response to each comment indicated by the reviewers, which are all contained in the attached “Response to the Reviewers”. We also have highlighted the changes in the text with red color for the ease of your review.
Lastly, we would like to thank you once again for providing us the opportunity to improve our manuscript.
Lastly, we would like to thank you once again for providing us the opportunity to improve our manuscript.
Sincerely yours,
Guan-Jhong Huang, Ph. D
School of Chinese Pharmaceutical Sciences and Chinese Medicine Resources,
College of Chinese Medicine, China Medical University, Taichung 404, Taiwan
E-mail: [email protected]
Point-by-point Response to the Reviewers
Reviewers' 1 comments:
Comments and Suggestions for Authors
The manuscript “Renoprotective effect of Pediococcus acidilactici GKA4 on cisplatin-induced acute kidney injury by mitigating Inflammation and Oxidative Stress and Regulating MAPK, AMPK/SIRT1/NF-κB, and PI3K/AKT Pathway” by Wen-Hsin Lin et al. They have; reported that Pediococcus acidilactici GKA4 could play an important role in the nephroprotection against cisplatin. The manuscript is written in good English language with few grammatical errors. Based on the current status of this manuscript, I think a major revision is necessary.
Comments:
- The authors should check grammatical errors very carefully before resubmission.
[Reply]: Thank you for your advice. As suggested, the revised manuscript has been read and corrected professionally by MDPI English language editor.
- In the introduction lines, no 1-11 have not been cited properly, I will suggest the authors cite these lines with recent references and add about acidilactici Grape King A4 strain (GKA4) with recent references in line no 41-46.
[Reply]: Thank you for your advice. As suggested, we changed recent references in the page 15, reference 1-2. In addition, we added the P. acidilactici Grape King A4 strain (GKA4) with recent references in the page 16, reference 16.
- Section 2.4. The research Design is confusing…. like on day 7, AKI was induced in mice by an intraperitoneal injection of cisplatin (20 mg/kg) in the cisplatin and GKA4 groups. The mice were sacrificed on day ten after cisplatin injection…it is confusing? Did the authors perform GKA4 and wortmannin at two different times? If not then why control and cisplatin groups have been repeated?
[Reply]: Thank you for your advice. As suggested, we more described it on page 3, line 26-30. The mice were administered GKA4 by oral gavage at doses of 62.5, 125 and 250 mg/kg once daily for 10 days (7 days before and 3 days after cisplatin injection). The control mice were orally administered saline daily. On day 7, AKI was induced in mice by an intraperitoneal injection of cisplatin (20 mg/kg) in the cisplatin and GKA4 groups. Mice were anaesthetized at 72 h after cisplatin injection to collect blood samples for measuring the serum biomarker, which were stored at -20 °C (1-3). In addition, The GKA4 and wortmannin groups were induced at two different times. Repeat the control and cisplatin groups as readers are better at understanding differences and also can serve as experimental controls.
Reference
- Zaaba NE, Beegam S, Elzaki O, Yasin J, Nemmar BM, Ali BH, Adeghate E, Nemmar A. The Nephroprotective Effects of α-Bisabolol in Cisplatin-Induced Acute Kidney Injury in Mice. Biomedicines. 2022, 10(4):842.
- Deng JS, Jiang WP, Chen CC, Lee LY, Li PY, Huang WC, Liao JC, Chen HY, Huang SS, Huang GJ. Cordyceps cicadae Mycelia Ameliorate Cisplatin-Induced Acute Kidney Injury by Suppressing the TLR4/NF-κB/MAPK and Activating the HO-1/Nrf2 and Sirt-1/AMPK Pathways in Mice. Oxid Med Cell Longev. 2020, 2020, 7912763.
- Mi XJ, Hou JG, Wang Z, Han Y, Ren S, Hu JN, Chen C, Li W. The protective effects of maltol on cisplatin-induced nephrotoxicity through the AMPK-mediated PI3K/Akt and p53 signaling pathways. Sci Rep. 2018, 8(1), 15922.
- In section 2.12. In western Blot Analysis, authors should add the specific primary and secondary antibodies along with their dilutions.
[Reply]: Thank you for your advice. As suggested, we added it on page 3, line 8-14.
- On page 13 line no 11 “AKI has a mortality rate of up to 50%” authors have mentioned there is 50% due to AKI, then why only 5 animals were taken in each group? Did the authors notice any mortality? Should be added to the manuscript.
[Reply]: Thank you for your advice. Mortality from AKI is as high as 20% and may be as high as 50% in the intensive care unit (ICU) (1). In animal models, the dose of cisplatin affects the degree of kidney damage. Current research on cisplatin-induced AKI mainly utilizes two mouse models, the short-term high-dose and long-term low-dose mouse models of nephrotoxicity. For example, long-term models use 5-15 mg/kg of cisplatin administered 2-4 times for 3-4 weeks. Short-term models use a single high dose of 20-30 mg/kg of cisplatin, which causes mortality and nephrotoxicity 3-7 days after cisplatin-induced AKI [2]. In this paper, the mice were administered GKA4 by oral gavage at doses of 62.5, 125 and 250 mg/kg once daily for 10 days (7 days before and 3 days after cisplatin injection). The control mice were orally administered saline daily. On day 7, AKI was induced in mice by an intraperitoneal injection of cisplatin (20 mg/kg) in the cisplatin and GKA4 groups.
In addition, only 5 animals were taken in each group because the “3Rs alternatives” refers to the Reduction, Refinement, and Replacement of animals used in research. The entire experimental protocol was approved by the Animal Protection Committee of China Medical University (CMUIACUC-2020-327). A key principle governing the ethical use of animals in research, testing and teaching is that no animal life is wasted; the number of animals used in each project must be the minimum necessary to obtain valid and meaningful results [3].
Reference:
- Luo M, Yang Y, Xu J, Cheng W, Li XW, Tang MM, Liu H, Liu FY, Duan SB. A new scoring model for the prediction of mortality in patients with acute kidney injury. Sci Rep. 2017, 7(1):7862.
- Holditch SJ, Brown CN, Lombardi AM, Nguyen KN, Edelstein CL. Recent Advances in Models, Mechanisms, Biomarkers, and Interventions in Cisplatin-Induced Acute Kidney Injury. Int J Mol Sci. 2019, 20(12), 3011.
- Public Health Service. (1996) U.S. Government Principles for the Utilization and Care of Vertebrate Animals Used in Testing, Research and training. PHS Policy on Humane Care and Use of Laboratory Animals. Washington, D.C.
- Representation of statistical notation (a..b….c.d ) is very difficult to understand. Explain statically significance in a simple but explanatory and easily understandable way.
[Reply]: Thank you for your advice. As suggested, we change the representation of statistics in the graph.
- Did the authors perform only a single set of protein expressions by western blot? Without statics. Results can not be trusted, so, multiple sets of blots must be performed, however, only a single set can be used in the manuscript but, an average of all sets must be mentioned.
[Reply]: Thank you for your advice. As suggested, all western blot experiments were repeated at least 3 times and the results were statistically analyzed in this paper, but only a representative image is shown. We had described it in the figure legends. In addition, we add the mean of all groups to our graph.

Reviewer 2 Report
The manuscript from Wen-Hsin Lin et al. describes the putative ability of Pediococcus acidilactici GKA4 to prevent kidney damages induced by antitumoral drugs. In that aim, the authors performed a series of experiments on animal models pre-treated with P. acidilactici GKA4 and then with cis-platin under several experimental conditions; different markers were then measured to evaluate the kidney functionality or damage.
The research is interesting and the reported data are promising, however, in my opinion two main limitations need to be modified.
First, there are numerous errors scattered throughout the text: figures inserted in the wrong order or not mentioned in the text, incomplete or meaningless sentences. in its present form it is not easy to read the manuscript and understand what the authors have done.
Second, many probiotics significantly affect the bioavailability of drugs, thereby reducing their effectiveness and side effects. In all the experiments performed by the authors, the drugs were administered a few days after the start of the animal treatment with P. acidilactici GKA4. Under these conditions, it is mandatory to measure whether the plasma levels of cis-platinum are comparable in animals treated differently. Without this information it is impossible to assess whether the observed effects depend on real protection from renal damage or on a reduction in the absorption of cis-platinum
Author Response
Point-by-point Response to the Reviewers
Reviewers' 2 comments:
Comments and Suggestions for Authors
The manuscript from Wen-Hsin Lin et al. describes the putative ability of Pediococcus acidilactici GKA4 to prevent kidney damages induced by antitumoral drugs. In that aim, the authors performed a series of experiments on animal models pre-treated with P. acidilactici GKA4 and then with cis-platin under several experimental conditions; different markers were then measured to evaluate the kidney functionality or damage.
The research is interesting and the reported data are promising, however, in my opinion two main limitations need to be modified.
First, there are numerous errors scattered throughout the text: figures inserted in the wrong order or not mentioned in the text, incomplete or meaningless sentences. in its present form it is not easy to read the manuscript and understand what the authors have done.
[Reply]: Thank you for your advice. As suggested, the revised manuscript has been read and corrected professionally by MDPI English language editor.
Second, many probiotics significantly affect the bioavailability of drugs, thereby reducing their effectiveness and side effects. In all the experiments performed by the authors, the drugs were administered a few days after the start of the animal treatment with P. acidilactici GKA4. Under these conditions, it is mandatory to measure whether the plasma levels of cis-platinum are comparable in animals treated differently. Without this information it is impossible to assess whether the observed effects depend on real protection from renal damage or on a reduction in the absorption of cis-platinum
[Reply]: Thank you for your advice. In animal models, the dose of cisplatin affects the degree of kidney damage. Current research on cisplatin-induced AKI mainly utilizes two mouse models, the short-term high-dose and long-term low-dose mouse models of nephrotoxicity. For example, long-term models use 5-15 mg/kg of cisplatin administered 2-4 times for 3-4 weeks. Short-term models use a single high dose of 20-30 mg/kg of cisplatin, which causes mortality and nephrotoxicity 3-7 days after cisplatin-induced AKI [1]. In this paper, the mice were administered GKA4 by oral gavage at doses of 62.5, 125 and 250 mg/kg once daily for 10 days (7 days before and 3 days after cisplatin injection). The control mice were orally administered saline daily. On day 7, AKI was induced in mice by an intraperitoneal injection of cisplatin (20 mg/kg) in the cisplatin and GKA4 groups.
Clinically, acute kidney injury caused by cisplatin usually occurs ten days after initiation of cisplatin treatment, and its clinical manifestations include decreased glomerular filtration rate, increased serum creatinine, and BUN. The mechanism of nephrotoxicity caused by cisplatin has been one of the main goals of scientific exploration. The current study found that when renal tubular cells are exposed to cisplatin, a series of reactions are initiated that eventually lead to renal tubular cell damage and even death. die. The inflammatory response caused by cisplatin also aggravates the damage of renal tubular cells. The renal vascular circulatory system will also be damaged by cisplatin and cause ischemic damage due to reduced blood flow to the kidneys. Combining the damage of various renal parenchyma described above, the nephrotoxicity of cisplatin leads to the final result of acute kidney injury [2].
In some scientific paper had shown that cisplatin-induced AKI produces an endogenous metabolic profile revealed by 1H-NMR analysis of urine. The most marked changes induced by cisplatin, and revealed by NMR spectroscopy, occurred within the first 48 h, and were characterized by increased concentrations of glucose, lactate, amino acids such as alanine, valine, leucine, methionine, and the presence of trichloroacetic acid cycle metabolites such as pyruvate and lactate in urine [3]. In addition, lactic acid bacteria (LAB) and their probio-active cellular substances exert many beneficial effects in the gastrointestinal tract. LAB prevent adherence, establishment, and replication of several enteric mucosal pathogens through several antimicrobial mechanisms. LAB also release various enzymes into the intestinal lumen and exert potential synergistic effects on digestion and alleviate symptoms of intestinal malabsorption (4). In addition, Pediococcus GKA4 was significantly improved in weight loss, colon length, disease activity index, and other indicators when compared with the negative control group. Pediococcus groups reduced the levels of pro-inflammatory cytokines IL-1β, IL-6, and TNF-α in serum; thereby, alleviating the intestinal inflammation caused by dextran sulfate sodium (DSS)-induced enteritis in mice (5). Based on these results, P. acidilactici GKA4 regulate the inflammatory effects and oxidative stress after cisplatin-treated AKI. Thus, this experiment uses mice to induce AKI, and plasma has been used to analyze inflammatory effects and oxidative stress. Plasma levels of cisplatin in animals with different treatments are currently not available. Therefore, we will further analyze its differences in the future.
Reference:
- Holditch SJ, Brown CN, Lombardi AM, Nguyen KN, Edelstein CL. Recent Advances in Models, Mechanisms, Biomarkers, and Interventions in Cisplatin-Induced Acute Kidney Injury. Int J Mol Sci. 2019, 20(12), 3011.
- Soni H, Kaminski D, Gangaraju R, Adebiyi A. Cisplatin-induced oxidative stress stimulates renal Fas ligand shedding. Ren Fail. 2018, 40(1):314-322.
- Portilla D, Li S, Nagothu KK, Megyesi J, Kaissling B, Schnackenberg L, Safirstein RL, Beger RD. Metabolomic study of cisplatin-induced nephrotoxicity. Kidney Int. 2006, 69(12):2194-204.
- Pessione E. Lactic acid bacteria contribution to gut microbiota complexity: lights and shadows. Front Cell Infect Microbiol. 2012, 2:86.
- Chen, Y.J.; Wang, C.S.; Tsai, Y.S.; Lin, S.W.; Wu, W.S.; Chen, Y.L.; C.C. Chen. Screening and evaluation of probiotics for reducing intestinal inflammation. Hans J. Food Nutr. Sci. 2022, 11(1), 44-55.

Round 2
Reviewer 1 Report
The authors have made significant improvements in the manuscript and have answered my comments in a scientific way with reference, so the manuscript can be accepted in its present form.
Reviewer 2 Report
in the new version of the manuscript some errors present in the previous one have been corrected. however, no changes have been made in relation to my main criticism of the manuscript: although the authors in their response describe further data regarding the harmful effects of cis-platinum and the benefits of the treatment with P. acidylactici GKA4, they did not evaluate the possible effects of the probiotic on the bioavailability of the drug. it would have been different if the authors had performed the probiotic treatment after inducing AKI with cis-platinum. But the experimental model described in the manuscript is in fact a co-treatment and it is therefore necessary to verify whether the simultaneous administration of the probiotic and the drug has effects on the plasma levels of the latter.
without data relating to this point the manuscript cannot be accepted for me.